# Heterogeneous Genomic Divergence Landscape in Two Commercially Important European Scallop Species

**DOI:** 10.3390/genes14010014

**Published:** 2022-12-21

**Authors:** David L. J. Vendrami, Joseph I. Hoffman, Craig S. Wilding

**Affiliations:** 1Department of Animal Behaviour, University of Bielefeld, Postfach 100131, 33615 Bielefeld, Germany; 2British Antarctic Survey, High Cross, Madingley Road, Cambridge CB3 OET, UK; 3School of Biological and Environmental Sciences, Liverpool John Moores University, Byrom Street, Liverpool L3 3AF, UK

**Keywords:** *Pecten*, scallop, RAD sequencing, genome differentiation, speciation, chromosomal inversion, inversion

## Abstract

Two commercially important scallop species of the genus *Pecten* are found in Europe: the north Atlantic *Pecten maximus* and the Mediterranean *Pecten jacobaeus* whose distributions abut at the Almeria–Orán front. Whilst previous studies have quantified genetic divergence between these species, the pattern of differentiation along the *Pecten* genome is unknown. Here, we mapped RADseq data from 235 *P. maximus* and 27 *P. jacobaeus* to a chromosome-level reference genome, finding a heterogeneous landscape of genomic differentiation. Highly divergent genomic regions were identified across 14 chromosomes, while the remaining five showed little differentiation. Demographic and comparative genomics analyses suggest that this pattern resulted from an initial extended period of isolation, which promoted divergence, followed by differential gene flow across the genome during secondary contact. Single nucleotide polymorphisms present within highly divergent genomic regions were located in areas of low recombination and contrasting patterns of LD decay were found between the two species, hinting at the presence of chromosomal inversions in *P. jacobaeus*. Functional annotations revealed that highly differentiated regions were enriched for immune-related processes and mRNA modification. While future work is necessary to characterize structural differences, this study provides new insights into the speciation genomics of *P. maximus* and *P. jacobaeus*.

## 1. Introduction

How genomes diverge during speciation is a central question in evolutionary genetics. Speciation is generally considered as a continuum where separate genetic lineages accumulate differences over time after a vicariant event due to mutation, genetic drift, and selection [1,2]. During periods of prolonged geographical isolation, genomic divergence can build up simply as a consequence of genetic drift. Additionally, when two separating lineages are exposed to different selective pressures, differentiation may be promoted by divergent selection [3]. Under this scenario, new beneficial mutations may undergo strong positive selection, which will increase their frequency and decrease genetic diversity at nearby sites via selective sweeps [4,5], further promoting divergence. In contrast, gene flow counteracts genetic divergence by homogenizing allele frequencies between diverging lineages. However, gene flow may not influence the entire genome equally, as chromosomal rearrangements such as inversions may favour reproductive isolation [6]. Notably, these areas are often hotspots for the accumulation of adaptive divergence [7]. Thus, the opposing forces of selection, genetic drift, and gene flow can give rise to heterogeneous landscapes of genomic divergence between separating taxa, with elevated differentiation at loci underlying divergent selection and loci involved in reproductive isolation [8,9]. Disentangling the mechanisms that led to such heterogeneous patterns is not trivial especially because apparently equivalent landscapes of genomic differentiation may arise from markedly different demographic histories, such as ecological divergence with continuous gene flow and allopatric divergence followed by secondary contact [8].

Relative measures of genetic divergence, such as *F*_ST_ [10], are sensitive to within-species levels of genetic diversity. Therefore, when using these kinds of measures, peaks of elevated genomic divergence along an heterogeneous landscape of differentiation will be identified both when these derive from differential gene flow across the genome, and also when these are the result of differential selection, even in the absence of gene flow [11]. This is because linked selection will reduce genetic diversity locally, resulting in unusually high estimates of divergence [11]. In contrast, absolute measures of genetic divergence, such as *d*_xy_ [12], are independent of levels of genetic diversity and will therefore identify peaks of divergence only in the presence of differential gene flow, as local reductions in genetic diversity caused by selection will not affect these kinds of estimates. Cruickshank and Hahn [11] proposed the comparison of relative and absolute measures of genetic divergence as an effective means of discriminating between the effects of differential selection and migration, while still identifying genomic regions potentially involved in adaptive divergence. This approach, together with demographic inference, should in principle allow the detailed reconstruction of the events and mechanisms responsible for generating heterogeneous landscapes of divergence during the speciation process.

The commercially exploited scallops *Pecten maximus* and *P. jacobaeus* are two Pectinids inhabiting different parts of the coast of Europe. *P. maximus* has a north Atlantic distribution and can be found from Norway to the western Mediterranean [13]. In contrast, *P. jacobaeus* is restricted entirely to the part of the Mediterranean that lies to the east of the Almeria–Orán front (AOF), a recognized biogeographic boundary [14] known to represent a major barrier for gene flow in both sessile species with planktonic larval phases (e.g., the urchin *Paracentrotus lividus* [15], the clam *Donax trunculus* [16], the mussel *Mytilus galloprovincialis* [17], the snail *Stramonita haemastoma* [18]), and species with actively swimming larvae and juveniles (e.g., *Dicentrarchus labrax* [19]). However, the extent to which the AOF is preventing and has prevented gene flow between *P. maximus* and *P. jacobaeus* remains unclear.

These two scallop taxa are currently considered as two separate species on the basis of morphological traits and nonoverlapping geographical distributions, but this view has been put into question by a suite of genetic studies published over the last two decades with different levels of genetic resolution offered by a variety of genotyping approaches including allozymes [13,20], mitochondrial DNA sequencing [20], microsatellites [21] and single nucleotide polymorphisms (SNPs) [22]. None of these studies identified any fixed differences between the two species and all of them reported genetic distances well below the magnitude expected under the original supposition that *P. maximus* and *P. jacobaeus* separated around five million years ago (MYA), shortly after the Messinian salinity crisis [23].

Vendrami et al. [22] examined genomic divergence within and between *P. maximus* samples collected from 12 sites along the Atlantic coast of Europe and *P. jacobaeus* samples collected from two sites along the Mediterranean coast. Clear genetic differences were found between *P. jacobaeus* and *P. maximus* yet the two species were estimated to have diverged much earlier than originally thought, sometime around the end of the Pleistocene (~95,000 generations or 0.5 MYA). Climatic conditions strongly fluctuated during that period and many bivalve species went extinct or shifted their distributions [24]. It is therefore reasonable to infer that newly established populations of the common ancestor of these two *Pecten* species might have begun to experience novel environmental pressures, leading them to diverge from populations located in their native range. Moreover, it is likely that divergence was also promoted by genetic drift and the consequent potential increase in reproductive isolation caused by the rise, for example, of Bateson–Dobzhansky–Muller incompatibilities [25,26,27,28]. Furthermore, Vendrami et al. [22], in line with the findings of Morvezen et al. [21], showed that *P. maximus* is composed of two major genetic lineages, one located along the coast of Norway and the other along the rest of the Atlantic coast of Europe, which diverged more recently during the last glacial maximum (~18,000 years ago). This finding was recently corroborated by Hollenbeck et al. [29] who also showed that chromosomal inversions appear to play an important role in maintaining adaptive variation in *P. maximus* populations inhabiting different regions along a European latitudinal gradient.

Recently, a high-quality chromosome level genome assembly for *P. maximus* was published by Kenny et al. [30] and this was subsequently refined by Zeng et al. [31]. Here we have reanalysed the RADseq data produced by Vendrami et al. [22] by mapping them to the reference genome of Zeng et al. [31] to characterize the genomic landscape of the divergence between *P. maximus* and *P. jacobaeus* and investigate its likely determinants. Our aims were, (i) to compare a series of demographic models of speciation to understand which are the mostly likely circumstances under which the divergence between *P. maximus* and *P. jacobaeus* occurred; (ii) to characterize and quantify relative and absolute divergence along the genome while also identifying, if present, any fully diagnostic loci; and (iii) to characterize the regions of these species’ genomes showing the highest levels of differentiation, as these may underlie important adaptive divergence.

## 2. Materials and Methods

### 2.1. RAD Sequencing Data

The RAD sequencing data originally described by Vendrami et al. [22] were downloaded from the Sequence Read Archive (SRA) using the accession number PRJNA526674. These included sequencing data from all 280 samples in Vendrami et al. [22], comprising both great scallops (*P. maximus*), collected from 12 locations from the Atlantic coast of Europe, and Mediterranean scallops (*P. jacobaeus*), collected from two locations in the Mediterranean Sea (Figure 1, Appendix A). Briefly, the whole genomic DNA was extracted from each sample from the adductor muscle tissue using an adapted phenol–chloroform protocol [32] and shipped to the Beijing Genomics Institute for RAD sequencing [33]. Libraries were constructed using the restriction enzyme PstI and 50 bp SE sequenced on an Illumina HiSeq 4000 (Illumina Hayward, Hayward, CA, USA).

### 2.2. Raw Read Processing, SNP Calling, and Genotyping

Raw read quality was assessed using FastQC v0.11.9 (http://www.bioinformatics.babraham.ac.uk/projects/fastqc/, accessed on 17 December 2021), which revealed that trimming was unnecessary as read quality scores were uniform along the entire length of the reads. Sequencing reads were then mapped to the *P. maximus* reference genome [31] using BWA MEM v0.7.13 [34] with default parameters. The resulting alignment files were used as input within the ref_map.pl module of the Stacks pipeline [35] for SNP calling and genotyping. Next, raw genotypes were filtered to retain only SNP calls with genotype quality and depth of coverage greater than five using VCFTools v0.1.17 [36]. Subsequently, we retained only SNPs genotyped in at least 80% of the individuals and discarded samples with more than 80% of missing data using VCFTools. The resulting VCF file was converted into a folded joint site frequency spectrum (SFS) using easySFS (https://github.com/isaacovercast/easySFS, accessed on 28 November 2022) to be used for demographic modelling (see Section 2.3). We note that the resulting SFS contained information on rare variants since no minor allele frequency (MAF) filtering was implemented at this point. Following this, VCFTools was utilized to further filter our SNP dataset for downstream analyses to retain only variants with an MAF greater than 0.05. We did not implement a Hardy–Weinberg equilibrium filtering step, as this would have removed highly divergent and fully diagnostic loci. In addition to this dataset, we generated five alternative SNP datasets to explore how sensitive the inference of genomic divergence (see Section 2.4) was to different filtering strategies. Specifically, we produced a dataset where only SNPs with genotyping rates below 80% in both species (as opposed to a global 80% threshold) were excluded. Two additional SNP datasets were obtained by removing SNPs with MAF below 1% and by not implementing MAF filtering, respectively. Finally, the remaining two datasets were generated by excluding SNPs that showed significant departures from Hardy–Weinberg equilibrium (HWE), using a *p*-value threshold of 0.05, either in all of the sampled populations or only in half of the sampled populations.

### 2.3. Demographic Modelling

We implemented demographic analyses using the coalescent simulator *fastsimcoal2* [37] in combination with the joint folded SFS obtained from our data, prior to MAF filtering, to reconstruct the divergence history of *P. maximus* and *P. jacobaeus*. Specifically, we compared five alternative models of speciation that differ in the absence, presence and timing of gene flow. The first model (SI, Appendix A) represents a model of divergence without gene flow and consists of two genetic lineages, one corresponding to *P. maximus* and one corresponding to *P. jacobaeus*, which diverged *T*_S_ generations ago. Constant population size was assumed throughout the model. The second model (IM, Appendix A) represents a model of isolation with constant gene flow and differs from the SI model by allowing constant gene flow between the two lineages after divergence. The third model (AM, Appendix A) represents a model of ancestral gene flow followed by isolation. In this case, gene flow occurs immediately after divergence, but stops in the past, *T*_1_ generations ago. The fourth model (SC, Appendix A) represents a model of secondary contact where after an initial period of allopatric divergence, gene flow is established *T*_1_ generations ago and continues until the present. Finally, the fifth model (SCS, Appendix A) represents another model of secondary contact which differs from the SC model by interrupting the gene flow *T*_2_ generations before the present. For all models, in addition to *T*_S_, *T*_1_, and *T*_2_, we estimated the effective population size of the *P. maximus* lineage (*N*_pma_), of the *P. jacobaeus* lineage (*N*_pja_), and of the ancestral lineage that later split into the two *Pecten* species (*N*_anc_). Moreover, in all models except SI, we also estimated the gene flow as the proportion of migrants per generation from the *P. maximus* lineage to the *P. jacobaeus* lineage (*mmj*) and vice versa (*mjm*). *mmj* and *mjm* were estimated independently from one another to allow for the possibility of asymmetrical gene flow to occur. All priors for these analyses are given in Appendix A.

For each model, we implemented 50 independent *fastsimcoal2* runs with 100,000 simulations and 40 cycles of the likelihood maximization algorithm. The run associated with the highest maximum likelihood value for each model was then selected to compute Akaike’s information criteria (AIC) as described in Varin and Vidoni [38] to allow for model comparison. We then extracted parameter estimates from the model receiving the highest support and computed their 95% confidence intervals (CI) via parametric bootstrapping, as described by Excoffier et al. [37]. Specifically, we simulated 100 folded SFS based on the parameter estimates from the best model and used them to recalculate the model parameters using the same workflow described above.

### 2.4. Comparative Genomics

In order to evaluate the degree of genetic differentiation between Mediterranean and great scallops along their genomes, we calculated *F*_ST_ values within sliding windows using VCFTools. Specifically, a 100 kb window was used, as recommended by Hohenlohe et al. [39] for RAD sequencing data, and only the windows containing at least five SNPs were considered. This analysis was replicated for five additional SNP datasets (see Section 2.2) to assess how sensitive the pattern of genomic divergence between the two species was to different filtering criteria. Windows whose *F*_ST_ was above the 95th percentile were considered as highly divergent genomic regions. For comparison, we repeated the same analysis where *F*_ST_ values were calculated between the Atlantic and Norwegian genetic lineages within *P. maximus*. Next, separately for the two scallop species, we used VCFTools to quantify nucleotide diversity within the same sliding windows used to calculate *F*_ST_ and tested for the association between these two measures by performing a Spearman correlation test. Subsequently, we employed the software pixy [40], in combination with a VCF file which also included nonvariant sites, to calculate *d*_xy_ between the two scallop species within the same genomic windows described above to verify whether genomic regions characterized by elevated *F*_ST_ values also showed high divergence regardless of local levels of genetic diversity. The correlation between these two measures was assessed using a Spearman correlation test. Moreover, we compared highly divergent genomic windows against the remaining genomic areas at their levels of divergence quantified using both *F*_ST_ and *d*_xy._ Agreement between these two measures would imply differential gene flow across the *Pecten* genome, while disagreement would indicate that heterogeneous patterns of *F*_ST_ along the genome were primarily caused by differential selection [11]. Finally, we aimed at identifying fully diagnostic SNPs for the two scallop species. Specifically, we searched for SNPs that were fixed for the reference allele in great scallops and fixed for the alternative allele in Mediterranean scallops.

### 2.5. Linkage Disequilibrium

In order to evaluate how linkage disequilibrium (LD) decays with physical distance along the genome in *P. maximus* and *P. jacobaeus*, we calculated *r*^2^ values among all pairs of loci located within the same chromosome, separately for the two species, using PLINK [41]. LD decay was then visualized by fitting a nonlinear regression curve, where the expected value of *r*^2^ under drift recombination was expressed according to Hill and Weir [42]. Next, we repeated the same analysis but focusing exclusively on SNPs located within highly divergent genomic windows to verify whether loci located on highly differentiated areas of the *Pecten* genome were characterized by a different LD decay pattern. Subsequently, we employed the R package ‘LDheatmap’ [43] to generate LD heatmaps showing LD among all pairs of SNPs separately for each chromosome and separately for the two species. This was performed in order to graphically assess whether any genomic regions were characterized by increased levels of LD, which would be indicative of a locally reduced recombination.

### 2.6. Gene Annotations and Enrichment Analysis

In order to functionally characterize highly divergent genomic regions of the scallop genome, we focused on the SNPs located within windows whose *F*_ST_ values were above the 95th percentile as well as on diagnostic SNPs (from now on collectively referred to as ‘highly divergent SNPs’). First, we used the great scallop reference genome annotations provided by Zeng et al. [31] to annotate highly divergent SNPs based on the genes they were located in or on the closest gene. Second, we retrieved gene ontology (GO) terms for all highly divergent SNPs using the annotations provided by Zeng et al. [31]. For SNPs not located within a gene, we used the GO annotation of the closest gene. These were used to perform a GO enrichment analysis using the R package topGO [44] to investigate whether any specific biological processes were overrepresented among the genes containing, or located in close proximity to, highly divergent SNPs. This was implemented while using a background list of GO terms containing all of the GO-annotated genes of the great scallop. In order to account for hierarchical relationships and nonindependence among the GO terms, we used the “weight01” algorithm with a Fisher statistic. GO terms with *p*-value < 0.05 were considered significantly enriched. We then repeated the GO enrichment analysis considering only SNPs located within genes. Finally, we used the program SnpEff v5.1 [45], together with a custom annotation file built using the great scallop reference genome annotations, to categorize highly divergent SNPs into different mutation classes. Specifically, this procedure allowed us to verify whether nucleotide changes within genes were synonymous, nonsynonymous, or loss-of-function mutations.

## 3. Results

We used the RADseq data of Vendrami et al. [22] to characterize patterns of genomic divergence between the two European scallop species, *P. maximus* and *P. jacobaeus*. RAD sequencing produced a total of 823,811,871 50 bp SE sequence reads. Alignment rates to the *P. maximus* reference genome were high (>90%) for both scallop species. The aligned reads were used to call a total of 1,009,368 raw SNPs. Application of the filtering criteria described in the materials and methods resulted in a final dataset consisting of 45,704 SNPs genotyped in 262 samples, including 235 great scallops and 27 Mediterranean scallops.

### 3.1. Demographic Inference

We used the coalescent simulator *fastsimcoal2* together with the empirical folded joint SFS to implement demographic reconstructions and to infer the divergence history of *P. maximus* and *P. jacobaeus*. All of the models that included gene flow between the two lineages received higher support than the SI model (Appendix A). Among them, the best supported model was the secondary contact SCS model. The resulting parameters estimates together with their 95% CI are reported in Appendix A. *P. maximus* and *P. jacobaeus* were estimated to have diverged 299,001 (95% CI: 235,482–329,076) generations ago. After a relatively long period of initial isolation, secondary contact occurred 14,799 (95% CI: 14,020–20,850) generations ago, with gene flow continuing until 58 (95% CI: 20–286) generations before present. Gene flow occurred in both directions but was more pronounced from the *P. jacobaeus* lineage to the *P. maximus* lineage.

### 3.2. Genomic Divergence

We calculated *F*_ST_ values between *P. maximus* and *P. jacobaeus* within 4460 100 kb sliding windows containing at least five SNPs. The resulting *F*_ST_ values revealed a highly variable divergence landscape between the two scallop species, with *F*_ST_ values ranging from −0.013 to 0.967 across the genome (Figure 2a) and an overall mean *F*_ST_ value of 0.118 (Table 1). This pattern was highly consistent across differently filtered SNP datasets (Appendix A), implying that our results are rather robust. A total of 223 windows showed *F*_ST_ values above the 95th percentile (*F*_ST_ = 0.623). These were located on chromosomes 1–3, 7, 9–15, and 17–19, with chromosomes 2 and 10 carrying the largest numbers of highly divergent genomic regions (18.38% and 28.46% of the total number of sliding windows, respectively) and having the highest chromosome-wise mean *F*_ST_ values (Table 1). In contrast, chromosomes 4–6, 8, and 16, as well as the unplaced scaffolds, did not carry any windows with *F*_ST_ values above the 95th percentile (Table 1). This genomic divergence landscape contrasts with that obtained when comparing the Atlantic and Norwegian genetic lineages of *P. maximus* (Figure 2b). Here, the mean, maximum, and 95th percentile values of *F*_ST_ were considerably lower (mean *F*_ST_ = 0.03, maximum *F*_ST_ = 0.65, 95th percentile = 0.11) and we could retrieve only a handful of peaks of high divergence, which were localized within comparably short genomic regions.

### 3.3. F_ST_, Nucleotide Diversity, and d_xy_ Values

We calculated nucleotide diversity separately for the two scallop species within the same sliding windows used to calculated *F*_ST_ values. Genome-wide nucleotide diversity (π) was 1.8 × 10^−5^ in *P. maximus* and 1.69 × 10^−5^ in *P. jacobaeus* (per-chromosome measures are available in Appendix A). The two species showed fairly similar distributions of nucleotide diversity across loci, with *P. jacobaeus* showing a larger number of windows with lower π (Figure 3 and Appendix A). *F*_ST_ and π were significantly negatively correlated in both species (*P. maximus*: *ρ* = −0.06, *p* < 0.01; *P. jacobaeus*: *ρ* = −0.33, *p* < 0.01; Figure 3). This indicates that the most divergent genomic regions tended to be located within genomic regions characterized by lower genetic diversity. Nevertheless, high *F*_ST_ values do not appear to be a by-product of locally reduced genetic diversity, as *d*_xy_ values calculated for *P. maximus* and *P. jacobaeus* within the same genomic windows showed a comparable pattern. First, chromosomes showing the highest average values of *F*_ST_ were also characterized by the highest average *d*_xy_ values (Table 1). Second, *F*_ST_ values were overall significantly positively correlated with *d*_xy_ values (*ρ* = 0.29, *p* < 0.01). Third, highly divergent genomic windows, when compared to the remaining windows, were characterized by higher levels of divergence as quantified by both *F*_ST_ and *d*_xy_ (Appendix A). This suggests that the gene flow at highly divergent genomic windows was substantially reduced in comparison to the remaining genomic regions.

### 3.4. Diagnostic Loci

Comparison of the genotypes from the two scallop species uncovered a total of 62 (0.13%) fully diagnostic SNPs (Appendix A). Notably, 54 (87%) of them were located in a single genomic region spanning approximately 10 Mb of chromosome 11 (Table 1). The other diagnostic SNPs were located on chromosomes 2 (four SNPs), 10 (two SNPs), 14 (one SNP), and 18 (one SNP, Table 1). A fasta file containing 100 bp of flanking sequence on either side of each fully diagnostic SNP is available at: https://figshare.com/articles/online_resource/P_maximus_vs_P_jacobeus_fully_diagnostic_loci_flanking_sequences/21311916 (last accessed on 20 December 2022), while the genotypes characteristic of each species can be found in Appendix A.

### 3.5. Linkage Disequilibrium

The pattern of LD decay was quantified separately for the two scallop species using only those SNPs that mapped to the 19 *P. maximus* chromosomes. SNPs located within unplaced scaffolds were excluded from this analysis. In *P. maximus*, LD was found to decay rapidly, with r^2^ reaching the background level (r^2^ = 0.01) by around 2 kb (Appendix A). This pattern did not change notably when restricting the analysis to SNPs located in highly divergent genomic regions, although the background level of LD was higher (Appendix A). LD appeared to decay rather rapidly also in *P. jacobaeus*, although it showed a slightly higher background level (r^2^ = 0.015) that was reached more slowly, at around 3 kb (Appendix A). In contrast, when analysing only SNPs located within highly divergent genomic regions in *P. jacobaeus*, r^2^ did not reach the background level until over ~400 kb (Appendix A). This suggests that differences in chromosome structure may be present in this species in the proximity of highly differentiated regions.

Visualization of LD values calculated among all pairs of loci separately for all 19 chromosomes revealed that highly divergent genomic regions were located in areas of elevated LD, suggesting low local recombination (Figure 4 and Appendix A). Moreover, in *P. jacobaeus*, regions of low recombination were often accompanied by reduced genetic diversity (in the absence of polymorphism at a given SNP, PLINK returns a NA value for LD which is plotted in white in Figure 4 and Appendix A). Notably, chromosomes lacking highly divergent genomic windows were not characterized by any blocks of high LD (Appendix A). Finally, chromosomes 2 (terminal part, Figure 4), 17, and 19 (Appendix A) contained genomic regions that showed elevated LD only in *P. maximus*.

### 3.6. Gene Annotations and Enrichment Analysis

The 223 sliding windows whose *F*_ST_ values were equal to or greater than the 95th percentile contained a total of 1777 SNPs, which included all of the fully diagnostic SNPs. These were found within or in close proximity to 438 different genes, for which we retrieved GO annotation terms. A GO enrichment analysis carried out against a background list of GO terms containing all of the GO-annotated genes of the great scallop revealed a total of seven significantly enriched biological process GO terms (Table 2 and Appendix A). Repeating the analysis considering only highly divergent SNPs located within genes yielded very similar significantly enriched GO categories (Appendix A).

The snpEff analysis revealed that the majority of the 1777 SNPs located within highly divergent genomic regions were located in non-coding regions, with 61% being situated outside of genic regions, 30% being located within introns, and 2.3% in UTRs (Table 3). Equivalent results were obtained when considering the diagnostic SNPs alone, with 63%, 29%, and 6% of the SNPs being located outside genic regions, in introns, and in UTRs, respectively (Table 3). Of the 107 highly divergent SNPs located within exons, 67, 40, and 2 represented synonymous, missense, and loss of function mutations, respectively (Table 3). Only one diagnostic SNP was located within a gene, and this was a synonymous mutation (Table 3). A full report of the snpEff annotations is given in Appendix A.

## 4. Discussion

The extent of genetic divergence between *P. maximus* and *P. jacobaeus* has been previously assessed, but most studies have used a limited number of loci (e.g., [13,20,21]). Even for the single study based on RADseq data [22], it was not possible to locate highly divergent regions due to the absence at the time of a reference genome. Here, we reanalysed RADseq data from Vendrami et al. [22] and mapped them to a new chromosome-level reference genome [31]. We demonstrate that genome-wide divergence between the two scallop species is highly heterogeneous, being characterized by peaks of divergence scattered throughout the genome, a pattern which contrasts sharply with that obtained when comparing Atlantic and Norwegian lineages within *P. maximus*, which are known to have diverged much more recently [22]. However, five of the 19 chromosomes showed little to no divergence. By implementing demographic and comparative genomic analyses, we were able to demonstrate that this heterogeneous pattern probably resulted from an initial prolonged period of isolation after divergence followed by secondary contact where genomic regions not involved in reproductive isolation have likely been homogenized by gene flow.

Our demographic analysis suggests that *P. maximus* and *P. jacobaeus* diverged around 300,000 generations ago and remained isolated until approximately 15,000 generations ago when secondary contact occurred. Gene flow then appears to have continued until around 60 generations before the present. Given a generation time of two to five years in *P. maximus* [42] (https://www.marlin.ac.uk/species/detail/1398, last accessed on 8 December 2022) this would imply that the two lineages split between around 0.6 and 1.5 million years ago (MYA) and that secondary contact started approximately between 30,000 and 75,000 years ago and lasted until a few centuries before the present. This is consistent with previous studies that estimated the split between the two *Pecten* species to have occurred in the Pleistocene during a period of fluctuating climatic conditions [13,22,46] and suggests that secondary contact might have occurred during the latter part of the last glacial period.

Similar models of allopatric divergence followed by secondary contact have been described for other Atlantic–Mediterranean sister lineages, including mussels [47] and sea bass [48]. Furthermore, both of these studies reported divergence to have occurred during the second half of the Pleistocene and secondary contact to have taken place around the end of the last glacial period. Additionally, a model of allopatric divergence followed by secondary contact yielding comparable estimates for the timings of both divergence and secondary contact, was also reported when studying the divergence between coastal and marine ecotypes of European anchovy populations [49]. Together, these studies and the present work suggest that shifts in range distributions during the Pleistocene had an important role in promoting evolutionary diversification in marine species with Atlantic and Mediterranean distributions. Moreover, the end of the last glacial period appears to have corresponded with geological and oceanographical changes within this geographic area, which provided the opportunity for marine lineages that diverged during an initial phase of allopatry to re-establish gene flow.

We recognize that our demographic parameter estimates are somewhat different from those obtained by Vendrami et al. [22], even though they are based on the same RADseq dataset. This is not necessarily surprising, as the current analyses differ from those of Vendrami et al. [22] in several ways. For example, our SFS was produced from a SNP dataset obtained from a reference-based genotyping approach, and not from a de novo based approach. Moreover, easySFS can incorporate SNPs with missing genotypes when computing the SFS via the down projection procedure described by [50] and is therefore likely to be more accurate. In addition, the demographic scenarios evaluated by Vendrami et al. [22] did not include gene flow, so the divergence time between the two *Pecten* species may have been underestimated. Consequently, we believe that our current demographic analyses represent an improvement over those of Vendrami et al. [22].

Based on our demographic inference, it appears that *P. maximus* and *P. jacobaeus* went through a period of isolation that lasted for around 285,000 generations, corresponding to around 0.5–1.5 million years. This indicates that divergence between the two species accumulated over a period of time ~20 times longer than the duration of secondary contact. Given that ecological parameters such as temperature, salinity, and water density clearly differ across the AOF [51], it is likely that strong divergent selection for different alleles between the two species, as well as positive selection for new beneficial mutations, may have been the primary drivers of divergence. The observation that genomic regions showing the highest *F*_ST_ values were also characterized by lower levels of genetic variability is consistent with this, and is suggestive of the occurrence of selective sweeps [4,5].

Genetic incompatibilities may also have increased during the prolonged period of isolation. The fact that we obtained consistent patterns of divergence using both relative (*F*_ST_) and absolute (*d*_xy_) measures of genomic divergence provides evidence that differential gene flow occurred after secondary contact and supports the presence of reproductive barriers within the *Pecten* genome. This implies that the observed highly divergent genomic regions are not simply a by-product of locally reduced genetic diversity [11], but may also be the consequence of reduced gene flow, a feature that has also been uncovered in the sea bass genome when comparing Atlantic and Mediterranean lineages [48,52]. Taken together, our results suggest that the highly differentiated genomic regions between the two *Pecten* species likely accumulated important adaptive divergence and were reproductively isolated. As a consequence, gene flow during secondary contact could not homogenize allele frequencies at these regions, resulting in a mosaic of highly and lowly differentiated genomic regions.

Loci under divergent selection are known to accumulate in regions of low recombination [53] such as centromeres [54] and the sites of chromosomal rearrangements [7,55]. Among them, chromosomal inversions have been linked to adaptation in several marine organisms, such as the Atlantic cod (*Gadus morhua*; [55,56]), three spined stickleback (*Gasterosteus aculeatus*, [7]), rough periwinkle (*Littorina saxatilis* [57]), and, notably, the great scallop (*P. maximus* [29]). In the latter study, which used a different reference genome (that of Kenny et al. [30]) and sequencing approach (ddRADseq) than our study, the authors identified inversions on three chromosomes that were associated with adaptive divergence among *P. maximus* populations sampled from different latitudes within Europe. These results may help to explain why we observed three genomic regions characterized by low levels of recombination exclusively in *P. maximus* and not in *P. jacobaeus*. Specifically, it is possible that these genomic regions, located on chromosomes 2, 17, and 19, may be of adaptive importance only in *P. maximus*.

Here we focused on the comparison between *P. maximus* and *P. jacobaeus* and tested whether SNPs located in highly divergent regions exhibited higher levels of LD. We found that, whilst overall LD decays rapidly in both species, when only divergent SNPs are considered in *P. jacobaeus*, LD extends beyond 400 kbp, indicating the likely presence of structural variants, such as chromosomal inversions [58]. Consistent with this hypothesis, all of the highly divergent regions were located in genomic areas showing high levels of LD and therefore low recombination. Taken together, the results of our study and that of Hollenbeck et al. [29] suggest that chromosomal inversions, or more generally genomic areas characterized by low levels of recombination, are important in European *Pecten* to maintain adaptive variation at different scales, both within different *P. maximus* lineages and also between *Pecten* species, where they act as intrinsic barriers to gene flow and maintain locally adapted variation.

In support of the argument for locally adapted variation, we found that highly divergent genomic regions between the two species appeared to be enriched for several biological processes. Furthermore, the genes enriched for GO-terms were distributed throughout the genome, discounting the possibility of this signal being produced by tandem repeats of the same gene. Among them, we found the “MyD88-dependent toll-like receptor signalling pathway” which is involved in the innate immune response of nonmammal species [59], including the scallop *Chlamys farreri* [60]. The evolution of the immune system is known to have influenced speciation in teleost fishes [61], while parasitism has also been proposed to play a role in ecological speciation [62]. It is therefore possible that differences in the immunological challenges encountered within the contrasting environments where *P. maximus* and *P. jacobaeus* evolved might have shaped patterns of genetic divergence between these two species. Furthermore, toll-like receptors also participate in development [63,64] and have been demonstrated to play a role in larval development in *Drosophila* [65]. If this is also the case for *Pecten*, divergence at genes associated with this pathway could potentially promote reproductive isolation, as larvae of one species dispersing within the distribution range of the other, may not be adapted to survive and develop under contrasting environmental conditions.

Genes involved in RNA modification also appeared to be over-represented in our enrichment analysis (GO terms: “snRNA 3’-end processing” and ”RNA (guanine-N7)-methylation”). In recent years, mRNA modifications have been implicated as regulatory mechanisms that control gene expression [66], which is known to be important for adaptation to novel environments [67]. Additionally, enrichment for the term “Mo-molybdopterin cofactor biosynthetic process”, which relates to the biosynthesis of the molybdenum cofactor, the active part of all molybdenum-containing enzymes [68], suggests that differences in the availability of molybdenum within the geographical ranges of the two scallop species, if present, might also have played a role in the speciation process.

Similar enriched GO terms were obtained when repeating the GO enrichment analysis considering only SNPs located within genes, corroborating our conclusions. Whilst highly divergent loci are located in low-recombination genomic regions, such that these SNPs could potentially be associated with any gene in the region where recombination is restricted, the fact that the majority of SNPs are outside coding regions suggests that most of the identified loci are likely to have regulatory roles rather than producing changes at the protein level. Similar results were obtained by Pujolar et al. [69] when studying patterns of divergence between the American eel (*Anguilla rostrata*) and the European eel (*A. anguilla*), and by Jones et al. [7], who investigated genomic divergence between marine and freshwater threespine sticklebacks. These authors found that 95.7% and 83%, respectively of the observed divergence was present in noncoding regions. Furthermore, divergence at regulatory elements plays a key role in speciation by promoting hybrid dysfunction via negative interactions between alleles [70]. Regulatory divergence between species is in fact widespread and has also been documented in mice, birds, flies, yeast, and plants [71,72,73,74,75].

Our demographic analyses further suggest that there currently is no gene flow occurring between *P. maximus* and *P. jacobaeus*. Consistent with this, we are not aware of any studies that have identified hybrid individuals based upon genetic data and this is also true of the current study [22], although none of our sampled populations were geographically close to the AOF. Whilst *P. jacobaeus* and *P. maximus* hybrids can be produced artificially [76] and measures of fertility (e.g., the gonadotrophic index) suggest that both species are reproductively mature at a similar time [77,78], opportunities for cross-species fertilization are likely to be limited. This is because spawning in scallops is triggered by temperature [79] and waters on the two sides of the AOF differ by 1–2 °C [14]. Moreover, among the genes within highly divergent genomic regions, we identified serine and threonine kinases, which are upregulated in scallop gonads [80,81]. This hints at the possible presence of differences in the reproductive cycles of these two species. Together, these arguments support the outcome of our demographic inference, as they suggest that gene flow is unlikely to occur at the present moment.

## 5. Conclusions

RADseq data for the commercially exploited European scallop species *P. maximus* and *P. jacobaeus* revealed a highly heterogeneous genomic divergence landscape between the two species. Our analyses indicate that this pattern likely originated as the consequence of a long period of isolation after divergence, during which the two lineages accumulated both adaptive divergence and reproductive barriers, followed by secondary contact, during which only those genomic regions permeable to gene flow were homogenized. Highly divergent genomic regions were associated with reduced levels of genetic diversity, which is indicative of the occurrence of selective sweeps and is consistent with the notion that positive selection imposed by contrasting environmental conditions played an important role in determining the pattern of genomic divergence observed today. Moreover, our analyses indicate that highly divergent genomic regions are located in areas of low recombination, hinting at the possible presence of chromosomal inversions, which are known to be important for maintaining adaptive genetic variation and promoting reproductive isolation. In line with this, highly divergent genomic regions also showed elevated *d*_xy_ values indicating that differential gene flow across the genome occurred during secondary contact. Finally, the genes located within highly differentiated regions are related to immunity, development, mRNA modification, and the biosynthesis of molybdenum cofactors, with the majority of SNPs likely affecting regulatory elements. Taken together, these results provide new insights into the divergence history of these two scallop species, while also identifying specific genomic regions that potentially contribute to the maintenance of adaptively important variation.

## Figures and Tables

**Figure 1 genes-14-00014-f001:**
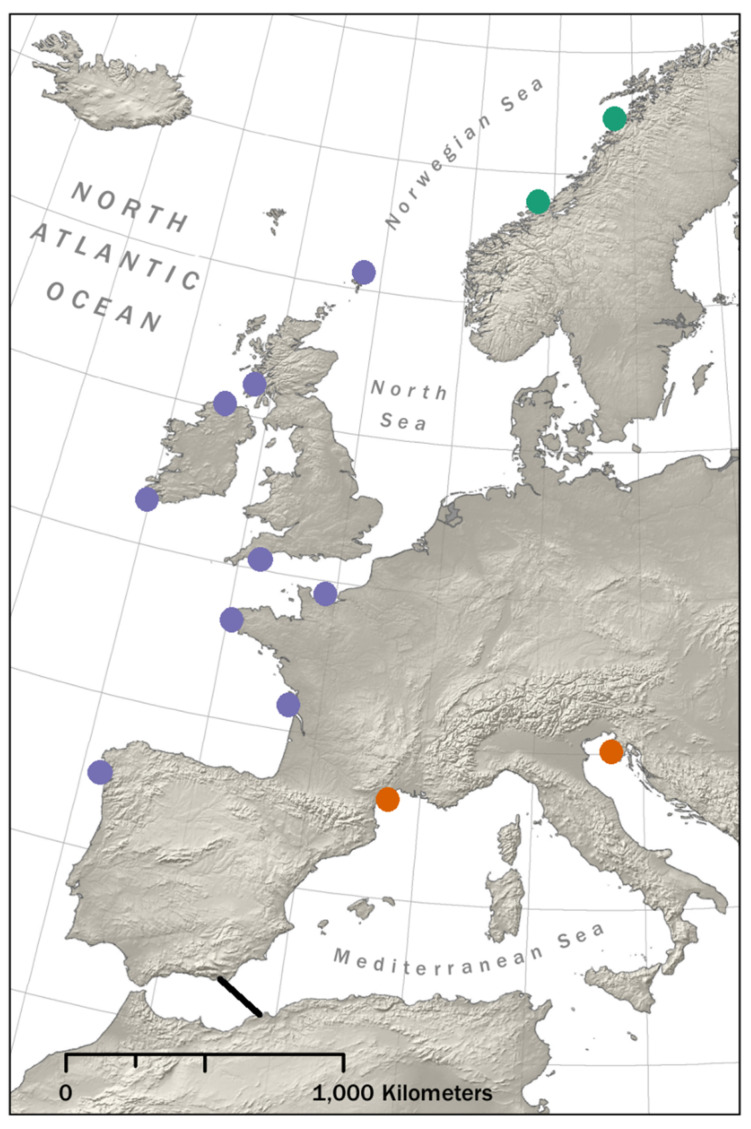
Map showing sampling sites (circles) and the location of the Almeria–Orán front (AOF, black line). Orange circles indicate sampling sites for *Pecten jacobaeus*, while the remaining circles indicate sampling sites for *P. maximus*. The *P. maximus* populations belonging to the previously identified Atlantic and Norwegian genetic clusters are indicated in purple and green [21,22], respectively.

**Figure 2 genes-14-00014-f002:**
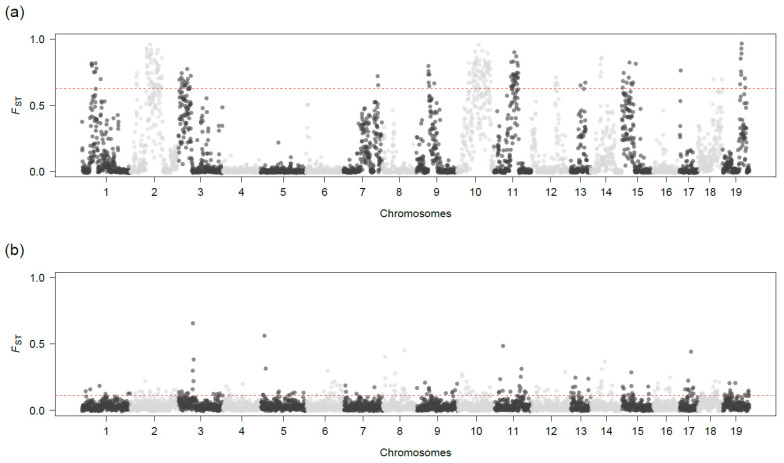
*F*_ST_ values calculated between samples within 4460 100 kb sliding windows. Panel (**a**) shows the comparison between *P. maximus* and *P. jacobaeus*, while panel (**b**) shows the comparison between Atlantic *P. maximus* and Norwegian *P. maximus*. Each point represents the *F*_ST_ value of a given window located on a given chromosome, as shown on the *x* axis. Light and dark grey blocks correspond to the 19 chromosomes. The red dashed lines represent thresholds corresponding to the 95th percentile *F*_ST_ values.

**Figure 3 genes-14-00014-f003:**
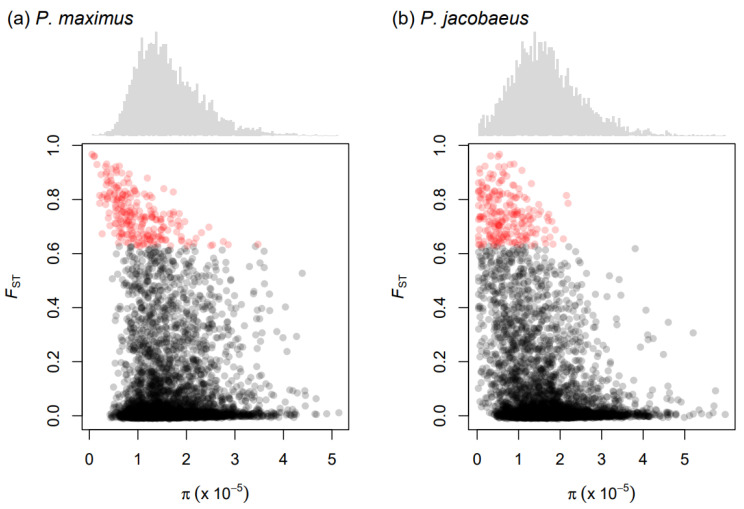
Relationship between nucleotide diversity (π) and *F*_ST_ values calculated in 100 kb sliding windows. Panels (**a**,**b**) shows results for *P. maximus* and *P. jacobaeus*, respectively. The histograms above the scatter plots show the distribution of nucleotide diversity calculated within sliding windows in the two scallop species. In the scatter plots, each point represents a sliding window. Red points indicate sliding windows with *F*_ST_ values equal to or greater than the 95th percentile.

**Figure 4 genes-14-00014-f004:**
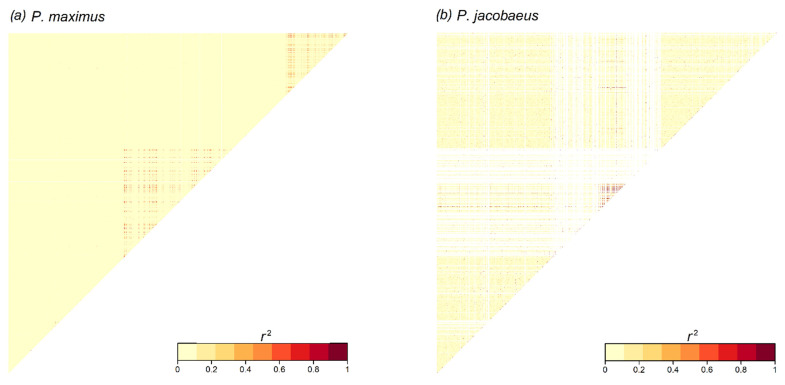
Heatmaps showing patterns of linkage disequilibrium (LD, *r*^2^) along chromosome 2, with darker areas corresponding to genomic regions characterized by elevated values of LD, as shown in the colour key. White lines refer to single nucleotide polymorphisms (SNPs) that were not polymorphic in a given species which resulted in a NA value when used to calculate LD. Panels (**a**,**b**) refer to *P. maximus* and *P. jacobaeus*, respectively. Red triangles highlight genomic regions characterized by higher levels of LD.

**Table 1 genes-14-00014-t001:** Number of sliding windows with an *F*_ST_ value above the 95th percentile (‘highly divergent’) by chromosome. The total number of sliding windows drawn within each chromosome (N) and the percentage of highly divergent windows (‘% highly divergent’) are shown together with the mean *F*_ST_ value and mean *d*_xy_ value. The number of diagnostic single nucleotide polymorphisms (SNPs) within each chromosome is also reported.

Chromosome	N	Highly Divergent	% Highly Divergent	Mean *F*_ST_	Mean *d*_xy_	Diagnostic SNPs
1	323	8	2.48	0.107	0.0045	0
2	321	59	18.38	0.259	0.0058	4
3	296	15	5.07	0.164	0.0051	0
4	253	0	0	0.007	0.0043	0
5	303	0	0	0.006	0.0040	0
6	250	0	0	0.017	0.0046	0
7	259	2	0.77	0.121	0.0046	0
8	228	0	0	0.031	0.0045	0
9	272	7	2.57	0.1	0.0045	0
10	246	70	28.46	0.398	0.0070	2
11	250	28	11.2	0.164	0.0054	54
12	262	5	1.91	0.079	0.0046	0
13	133	2	1.5	0.087	0.0046	0
14	209	5	2.39	0.108	0.0048	1
15	209	10	4.78	0.168	0.0058	0
16	176	0	0	0.02	0.0049	0
17	122	1	0.82	0.034	0.0047	0
18	166	2	1.21	0.169	0.0057	1
19	179	9	5.03	0.127	0.0057	0
Unplaced scaffolds	3	0	0	0.016	0.0049	0
Total	4460	223	5.0	0.118	0.0050	62

**Table 2 genes-14-00014-t002:** List of significantly enriched gene ontology (GO) terms relative to genes containing or located in the proximity of highly divergent SNPs.

GO Term ID	Description	*p*-Value
GO:0002755	MyD88-dependent toll-like receptor signalling pathway	<0.01
GO:0009058	biosynthetic process	<0.01
GO:0034472	snRNA 3’-end processing	<0.05
GO:0032456	endocytic recycling	<0.05
GO:0036265	RNA (guanine-N7)-methylation	<0.05
GO:0006777	Mo-molybdopterin cofactor biosynthetic process	<0.05
GO:0006465	signal peptide processing	<0.05

**Table 3 genes-14-00014-t003:** Details of the 1777 highly divergent SNPs, including the 62 fully diagnostic SNPs. For each type of SNP, we indicate whether the SNP is located within a gene and whether it is transcribed. We also indicate how many highly divergent SNPs and fully diagnostic SNPs are attributed to each mutation category. Proportions are reported in parenthesis.

SNP Type	Within Gene	Transcribed	Highly Divergent	Diagnostic
intergenic region	N	N	724 (0.41)	24 (0.39)
downstream gene variant	N	N	166 (0.09)	10 (0.16)
upstream gene variant	N	N	196 (0.11)	5 (0.08)
3’ UTR variant	Y	N	34 (0.02)	4 (0.06)
5’ UTR variant	Y	N	6 (0.003)	0 (0)
intron variant	Y	N	531 (0.3)	18 (0.29)
splice region variant and intron variant	Y	N	11 (0.01)	0 (0)
synonymous variant	Y	Y	67 (0.04)	1 (0.02)
missense variant	Y	Y	40 (0.02)	0 (0)
loss of function	Y	Y	2 (0.001)	0 (0)

## Data Availability

The sequencing data analysed here are available (SRA, accession number PRJNA526674) and were originally described by Vendrami et al. [22]. The code for this study is available at: https://github.com/DavidVendrami/Pmaxiums_jacobeus_genomic_divergence (last accessed on 20 December 2022).

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
