# Peer review of "Heterogeneous Genomic Divergence Landscape in Two Commercially Important European Scallop Species"

_genes, 2022, doi:10.3390/genes14010014_

Round 1

Reviewer 1 Report

I went through the manuscript by Vendrami et al. describing the presence of large islands of differentiation between two species of scallop inhabiting different side of the Almeria-Oran front. The analyze of GO-term shows an enrichment in immune-related gene, suggesting some adaptive divergence.

I find the system interesting. However, I think the manuscript is rather descriptive and does not bring much more to what is already know from the system. Moreover, the manuscript misses some key hypothesis/reference to study the evolution of divergence island, making some conclusion of the manuscript rather speculative at this stage. In addition, there is a step in the filtering procedure (the MAF of 5%) that is likely to bias the results with the unbalanced data analyzed here (235 individual of one species vs 27 of the other one). 

major comment:

1.       There is a long discussion on the origins of the heterogeneous landscape of divergence (See Ravinet et al. 2017 https://doi.org/10.1111/jeb.13047), notably to assess if this heterogeneity are the result of migration-selection balance during “recent ecological speciation with gene flow” or the result of prior isolation followed by secondary contact where region not involved in reproductive barrier have been homogenize by secondary gene flow. The manuscript completely misses this second interpretation.

2.       Similarly, the heterogeneous landscape of differentiation can arise in absence of gene flow due to the Robertson-Hill interference, which is expected to be strong in regions of low recombination. This is completely absent from the current manuscript. I recommend the authors to add reference to Cruickshanks and Hahn (2014 https://doi.org/10.1111/mec.12796). I think that the manuscript would benefits from adding some formal test to distinguish the role of Hill-Robertson interference from barrier to gene flow: see suggestion part.

There is some extensive work from on Sea Bass in the same area that have studied all this (e.g. https://www.nature.com/articles/ncomms6770, https://www.nature.com/articles/s41467-018-04963-6 )

3.       MAF 0.05 overall is not suited with such unbalanced sampling (232 individuals from one species vs 27 from the other one), as 5 % of 262 diploid individuals is ~26.This means that all SNPs with private allele (allele that are not differentially fixed but only found in one species) showing frequency of less than 0.50 in P. jacobeus will be removed. I expect them to be many with the extent of differentiation between both species here.

Overall, this tend to keep only SNPs that are either well differentiated between the two species (with a frequency differences > 0.5), or SNPs that are in shared frequency between the two species (SNPs that flow?), which in turn inflate the heterogeneity in landscape of

differentiation, as all SNPs with intermediate level of differentiation are removed.

Suggestion:

To explore the origins of the divergence island (secondary contact vs isolation with migration, barrier to gene flow vs roberston-hill effect) you could consider adding some demographic inferences. I recommend inference by likelihood approximation using one of the following software:

-          dadi https://dadi.readthedocs.io/en/latest/

-          moments https://bitbucket.org/simongravel/moments/src/main/

The model to test for different speciation scenario are available from Quentin Rougemont’ GitHub for dadi: https://github.com/QuentinRougemont/DemographicInference or from Paolo Momigliano’ Github for moments https://github.com/Nopaoli/Demographic-Modelling

minor comment

I am not sure what is the reason to keep all P. maximus samples? They just add noise to the purpose of the manuscript

Specific comment.

abstract: “indicative of recent ecological speciation with gene flow”: This is only one possible interpretation that is highly speculative at this stage, thus, this statement should be removed. Many case of secondary contact have been described between lineages from each side of the Almeria-Oran front (mussel, sea bass…). This hypothesis should be tested first before any further claims on ecological speciation.

l39: “may favor reproductive isolation at specific sites” sounds rather odd, rephrase or remove “at specific sites”

l74-76: This is true, but this isn’t the only things that happened during the fluctuation of climate. Indeed, the connectivity from each side of the AOF is reduced during the glacial period (sometime leading to complete isolation). This isolation favor divergence due to drift, but also incompatibilities like Bateson-Dobzanski-Muller to increase in frequency. Adaptation alone is highly unlikely to generate the signal detected here. I suggest the authors to mention the alternative interpretation here and rely more on the body of work conducted on the topic cited in the paragraph above

line 295: it’s sounds rather weird to start a discussion by “Whilst”

table 2: why not adding the number of diagnostic SNPs per chromosome here?

Figure 3: These heatmap need a scale normally to see how far apart are the SNPs on the chromosome. This can be done easily with LDheatmap.

Reviewer 2 Report

General comments

Vendrami et al. report a genomic study of two European scallop
species, revealing a heterogeneous pattern of differentiation across
the genome. The paper is well-written, and the study builds on recent work by the authors and others to characterize population genomics and structure of Pectinids in the north Atlantic, and also adds to a growing body of work implicating chromosomal inversion in adaptation and speciation.

One important thing to note is that the SNP filtering parameters used may be masking some important signal. When sample sizes for different groups are very uneven (P. jacobeus is <10% of the total samples) it is important to pay close attention to global filters for missing data and minor allele frequency, as was applied here. For missing data, removing SNPs only typed in 80% of individuals will remove any variation that is called only in P. jacobaeus (due to indels, for example) but not called in P. maximus. Given recent studies that have shown a high degree of indel variation in scallops and other bivalves (Calcino et al. 2021: https://doi.org/10.1098/rstb.2020.0153), this is worth evaluating. Throwing out these regions before the analysis would bias analysis to make the species seem more similar than they actually are. A plot of missing data proportion (P. maximus vs P. jacobaeus) by genome coordinates could reveal something here. The same principle applies to MAF: a fairly common allele in P. jacobaeus that was not present in P. maximus would get removed from the data under the current filtering, only because there are so many more P. maximus samples in the dataset.

My second major suggestion is that I would suggest modifying the discussion to include alternative hypotheses for the overall explanation of heterogeneous genomic divergence that is given in the paper (ecological speciation with gene flow). While areas of low recombination have been implicated in this model of speciation, this is not the only explanation, and alternative models include allopatric speciation followed by secondary contact without gene flow, where similar patterns of genomic divergence can be explained by heterogeneous patterns of genetic diversity and local adaptation without gene flow. These alternatives are hard to distinguish with relative measures of divergence like FST. See the specific comments for a reference that is a good starting point for this discussion.

Specific comments

Table 1: A map of the sampling locations and which indicates the biogeographic barrier(s) mentioned would be more helpful than Table 1, which would be fine to put in the supplemental materials.

Line 69: "populations" would be more accurately termed "samples" or something similar here

Line 122-123: You can implement a HWE filtering step, as long as you run the test for each discrete population/species.

Lines 63-64, 232-234, 296-297: Can you explain why the previous RAD study with this dataset was not able to find any diagnostic SNPs? A reference genome is not necessary to calculate allele frequencies.

Lines 240-250: Could differences in LD decay be attributed to genomic differences between the species? Here, distance is determined from the P. maximus genome, yet it seems possible that there are differences in chromosome structure, which would bias the estimates for P. jacobaeus. It is possible that analysis of genomic patterns of missing data (see general comments) between species would inform this.

Figure 3: Spelling of P. jacobaeus is inconsistent in the figure and in several places throughout the manuscript

Table 4: "Loss" should not be capitalized for consistency with other row labels

Discussion: while the discussion of hypotheses related to adaptive divergence is interesting, it should be noted that the fact that divergent loci are associated with low-recombination regions means that SNPs could be associated with any gene in the region where recombination is restricted, not just the one closest. Also, did you check to make sure that gene enrichment didn't have to do with tandem repeats of the same gene? In other words, were genes associated with enriched GO-terms adjacent to each other or distributed throughout the genome?

Lines 334-335: Do you mean that those chromosomes are labeled differently in the different reference genomes?

Line 366: Can you reword? This reads as if the mRNA modifications are emerging as a regulatory mechanism (meaning that they weren't operating in the past), but I assume you mean that mRNA modifications have been more often implicated as regulatory mechanisms recently.

Lines 407-410: the conclusion that heterogeneous genomic divergence is the product of ecologically-driven divergent selection in the face of gene flow seems at odds with the previous paragraph which proposes evidence for a lack of gene flow.

Given the use of FST rather than a measure of divergence that is not sensitive to levels of within-species diversity, some of the discussion should include the hypothesis that the heterogeneous divergence may not be a result of gene flow at all. See Cruickshank and Hahn 2014: doi: 10.1111/mec.12796 for discussion of this idea and some of the pitfalls of this type of analysis when dealing with FST and species-level divergence.

Round 2

Reviewer 1 Report

I have been through the revision by Vendramiand collaborators of the manuscript entitled "Heterogeneous genomic divergence landscape in two commercially important European scallop species" and the answer provided to my comments. 

I think the authors did a great job in adressing all of the comments provided by the reviewer. I feel that the manuscript has much improve and is also much more nuanced than before. I am thus happy to endorse the manuscript for publication. Congrat to the authors for this nice piece of work!

I only have one minor comment/issue: Based on the current manuscript, it is unclear on what dataset was performed the demographic inference. Normally, demographic inferences based on the jSFS require to keep all SNPs (even those with maf < 5%) as they carry relevant information about effective size (and perhaps history).  Removing them might lead to wrong inferences. Did you keep the rare variants prior to compute the jSFS?

I also found evidence for a secondary contact with similar estimates split and contact in the European anchovy : https://onlinelibrary.wiley.com/doi/full/10.1111/mec.13627. This manuscript could perhaps be relevant in the discussion in l458 :)
